# Efficacy and Safety of Rivaroxaban, Apixaban, and Edoxaban for Nonvalvular Atrial Fibrillation Based on Blood Coagulation Activity and Drug Plasma Concentration: SETtsu and North Osaka Multicenter Direct Oral AntiCoagulant (SET DOAC) Registry

**DOI:** 10.3390/ph17111431

**Published:** 2024-10-25

**Authors:** Michihiro Suwa, Isao Morii, Masaya Kino, Yumie Matsui, Masahiro Yoshinaga, Hiroki Takahashi, Masahiko Takagi, Akira Yoshida, Minoru Ichikawa, Osamu Nakajima, Mitsuhiro Tanimura, Hisashi Shimoyama, Hiroyuki Saitoh, Isao Sasaki, Takeshi Suzuki, Satoshi Uemae

**Affiliations:** 1Department of Cardiology, Hokusetsu General Hospital, Osaka 569-8585, Japan; 2Department of Cardiology, Saiseikai Izuo Hospital, Osaka 551-0032, Japanyoshinaga-lasso@ezweb.ne.jp (M.Y.); 3Department of Cardiology, Kansai Medical University Medical Center, Moriguchi 570-8507, Japan; 4Department of Cardiology, Higashiosaka Municipal Hospital, Higashiosaka, 578-8588, Japanmichikawa419@gmail.com (M.I.); 5Department of Cardiology, Hirakata Municipal Hospital, Hirakata 573-1013, Japan; haku-n@hera.eonet.ne.jp; 6Midorigaoka Hospital, Takatsuki 569-1121, Japan; mitsuhiro-tanimura@midorigaoka.or.jp; 7Itami City Hospital, Itami 664-8540, Japan; 8Department of Cardiology, Yukoukai General Hospital, Ibaraki 567-0058, Japan; hiro-saito@yukoukai.com; 9Ainomiyako Neurosurgery Hospital, Osaka 538-0044, Japan; yakuzaibu@ainomiyako.net; 10Sysmex Corporation, Kobe 651-2241, Japan

**Keywords:** anticoagulation monitoring, apixaban, atrial fibrillation, edoxaban, on-label dosing, rivaroxaban

## Abstract

**Background:** The therapeutic effects of oral anticoagulant drugs for nonvalvular atrial fibrillation (NVAF) suggest that the three factor Xa (FXa) inhibitors may have distinct safety profiles, though this is not yet fully conclusive. This study investigated the current dosing of rivaroxaban, apixaban, and edoxaban by monitoring drug plasma concentration (PC) and coagulation activity from the viewpoint of the safety. **Methods and results:** This multicenter clinical study monitored the drug PC and two coagulation biomarkers (fibrinogen and fibrin monomer complex [FMC]) at peak and trough timing in 268 outpatients taking rivaroxaban (n = 72), apixaban (n = 71), and edoxaban (n = 125) for NVAF. Doses were adjusted based on the dose-adjustment criteria of each drug. Referencing our previous study, peak drug PC remained below the cut-off level for predicting bleeding events except in eight patients (rivaroxaban, n = 3; apixaban, n = 2; edoxaban, n = 3) in whom bleeding events occurred. Among them, two (one each on rivaroxaban and edoxaban) had a peak drug PC below the cut-off level. Drug PCs widely varied from peak to trough, whereas FMC levels, reflecting thrombin activity, remained within the normal range (<6.1 µg/mL) regardless of PC variations. These results indicated that the anticoagulant effects of these drugs persisted throughout the day regardless of the drug PC levels, dosage, and dosing frequency. Regarding the change over time in peak PC, the elevation over time developed more in rivaroxaban (29/57; 50.9%, *p* < 0.05) than in edoxaban (32/101; 31.7%), and rivaroxaban tended to accumulate more than edoxaban. **Conclusions:** Although drug PC levels of once-daily FXa inhibitors widely varied from peak to trough, FMC levels were maintained within the normal range without daily variations. Rivaroxaban also tended to accumulate over time. The results indicate the low risk of thrombotic events with once-daily FXa inhibitors and its correspondence to the twice-daily regimen.

## 1. Introduction

The 2021 European Heart Rhythm Association (EHRA) practical guide on the use of direct oral anticoagulants (DOACs) in patients with nonvalvular atrial fibrillation (NVAF) specified that DOACs provide a similar degree of antithrombotic effects as vitamin K antagonists with fewer bleeding events [1]. However, rivaroxaban has a higher bleeding incidence than apixaban and edoxaban [2,3,4,5,6,7]. Additionally, in patients with low body weight (BW), two practical guides for DOAC use stated that individuals with a BW of ≤60 kg needed special care and that reduced doses of apixaban and edoxaban should be administered [1,8]. Furthermore, the EHRA guidelines stated that monitoring the plasma levels of DOAC is appropriate for individuals with a BW of <60 kg for whom rivaroxaban or dabigatran was prescribed. Furthermore, our previous work (Hokusetsu DOAC study) demonstrated that edoxaban exhibited a more favorable safety profile compared to rivaroxaban, based on plasma concentration (PC) monitoring [9]. So, this finding indicated that these three factor Xa (FXa) inhibitors may have distinct safety profiles. This study aimed to compare the efficacy and safety of three FXa inhibitors by measuring their PCs at peak and trough levels, assessing antithrombotic action through coagulation biomarkers, and evaluating the correlation between drug PCs over time, including daily variations, and anticoagulant activity (the SET DOAC registry; UMIN 000036769).

## 2. Results

### 2.1. Patient Characteristics (Table 1)

Among the 289 outpatients registered at all nine hospitals, 21 (rivaroxaban, n = 8; apixaban, n = 6; edoxaban, n = 7) were excluded because of the off-label low-dose use and/or withdrawal of consents. Table 1 shows the baseline characteristics of the 268 participants (rivaroxaban, n = 72; apixaban, n = 71; edoxaban, n = 125). More patients were using the standard dose in rivaroxaban (57/72, 79%) and apixaban (41/71, 58%) dosing, but more patients were taking the low dose in edoxaban (81/125, 65%). Expectedly, the BW and baseline creatinine clearance (Cr-Cl) were significantly lower in patients who received low doses than in those who received standard doses of rivaroxaban and edoxaban. The low-dose subgroup was significantly older than the standard-dose subgroup in all these three drugs. However, no differences in the baseline CHADS_2_ score, NVAF patterns, presence of hypertension (HT), and antiplatelet drug use were noted between the standard- and low-dose subgroups in all DOAC groups (Table 1).

**Table 1 pharmaceuticals-17-01431-t001:** Patient profiles of 3 FXa inhibitor users.

Drug	Rivaroxaban (n = 72)		Apixaban (n = 71)		Edoxaban (n = 125)	
dose	Standard 15 mg od	Low 10 mg od	Standard 5 mg bid	Low 2.5 mg bid	Standard 60 mg od	Low 30 mg od
	(n = 57)	(n = 15)	(n = 41)	(n = 30)	(n = 44)	(n = 81)
M/F	43/14	6/9	18/23	11/19	31/13	36/45
Age(yr)	65 ± 10	76 ± 8 (*)	73 ± 7	80 ± 7 (*)	68 ± 9	77 ± 7 (*)
BW(kg)	73 ± 14	60 ± 12 (*)	62 ± 10	56 ± 9	74 ± 13	55 ± 10 (*)
Baseline Cr-Cl (mL/min)	88.4 ± 32.6	42.8 ± 8.4 (*)	63.4±19.1	46.4 ± 18.9	82.7 ± 26.2	51.4 ± 16.1 (*)
CHADS2 scores	1.2 ± 0.8	2.1 ± 0.7	1.5 ± 1.2	2.0 ± 0.9	1.5 ± 1.1	1.8 ± 1.1
NVAF pattern paroxysmall/ permanent (n)	19/38	5/10	14/27	10/20	14/30	27/54
HT (+/−)	33/24	5/10	17/24	14/16	32/12	43/38
APT use (+/−)	6/51	1/14	2/39	2/28	3/41	6/75

APT: antiplatelet therapy; BW: body weight; Cr-Cl: creatinine clearance; HT: hypertension; M/F: male/female; NVAF: nonvalvular atrial fibrillation; (*): statistically significant difference; *p* < 0.05 (between standard and reduced doses of each DOAC).

### 2.2. Drug Peak and Trough PCs over Time in the Standard and Low-Dose Subgroups in All DOAC Groups (Figure 1)

Drug PCs sampled from the standard (on right) and low-dose (on left) users in all three drugs at peaks 1–3 and troughs 1 and 2 over time are shown in Figure 1 (rivaroxaban, top; apixaban, middle; and edoxaban, bottom).

**Figure 1 pharmaceuticals-17-01431-f001:**
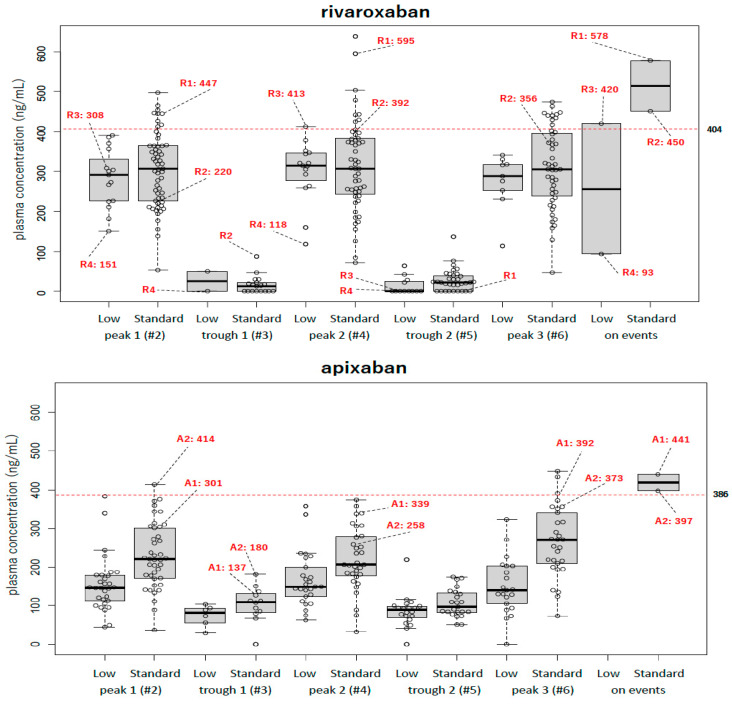
Alteration in peak and trough PCs over time in 3 FXa inhibitors. Alteration in peak (peaks 1–3) and trough (troughs 1 and 2) PCs over time, corresponding to numbers #2–#6 in flow chart in Figure 2. Bottom of box, 25th percentile; horizontal line, 50th percentile (median); top of box, 75th percentile; whiskers, maximum and minimum non-outliers, respectively. Red dotted lines indicate cut-off level for predicting bleeding events for each drug. Bleeding events in users of rivaroxaban, apixaban, and edoxaban are highlighted with red numbers for each PC category.

**Figure 2 pharmaceuticals-17-01431-f002:**
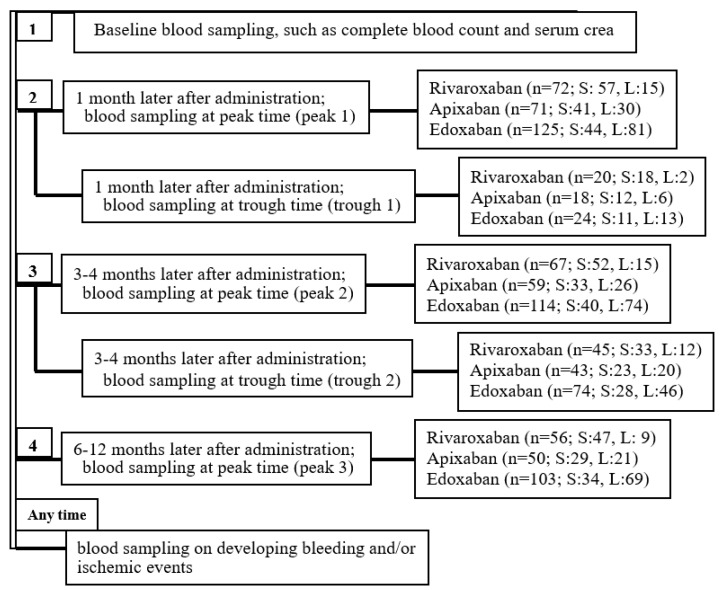
Flowchart of blood sampling (from the order of #1 to #6) and patient enrollment. S: standard-dose use, L: low-dose use.

In the standard-dose subgroup (on right), the median (IQR) peak PC values (ng/mL) of rivaroxaban users over time were 307.2 (225.3–364.0; n = 57), 307.4 (240.6–382.7; n = 52), and 305.8 (234.5–398.6; n = 47) at peaks 1, 2, and 3, respectively, which corresponded to #2, #4, and #6, shown in the flow chart of Figure 2. The median (IQR) peak PC (ng/mL) values of apixaban users over time were 221.0 (170.4–302.8; n = 41), 206.2 (175.6–280.3; n = 32), and 270.7 (204.6–345.9; n = 29) at peaks 1, 2, and 3, respectively. The median (IQR) peak PC (ng/mL) values of edoxaban users over time were 278.2 (210.3–331.7; n = 44), 240.6 (196.9–327.7; n = 40), and 271.7 (227.5–322.5; n = 34) at peaks 1, 2, and 3, respectively.

In the low-dose subgroup (on left), the median (IQR) peak PCs (ng/mL) of rivaroxaban users over time were 290.5 (224.3–356.5; n = 15), 314.2 (263.1–347.6; n = 15), and 287.7 (241.5–323.5; n = 9) at peaks 1, 2, and 3, respectively. For apixaban users, the median (IQR) peak PCs (ng/mL) over time were 146.6 (109–178.9; n = 30), 149.0 (120.7–205.0; n = 26), and 140.0 (99.4–202.5; n = 21) at peaks 1, 2, and 3, respectively. For edoxaban users, the peak PCs (ng/mL) over time were 173.6 (132.1–197.4; n = 81), 182.3 (148.6–221.1; n = 74), and 175.4 (137.5–209.8; n = 68) at peaks 1, 2, and 3, respectively.

For evaluating PC alteration over time, we compared peak PC data at the peak 1 time among these three FXa inhibitors but there was no significant difference in the standard-dose subgroup. In contrast, in the low-dose subgroup, the peak PC at the peak 1 time for rivaroxaban was significantly higher than those values for apixaban and edoxaban dosing (*p* < 0.05).

In the standard-dose subgroups, the median (IQR) trough PCs (ng/mL) of rivaroxaban users over time were 13.0 (0–23.5; n = 18) and 22.4 (0–41.4; n = 33) for troughs 1 and 2, respectively. The trough PCs of apixaban users were 110.2 (80.6–134.6; n = 12) and 96.8 (83.0–135.2; n = 23) at troughs 1 and 2, respectively. The trough PCs of edoxaban users were 19.0 (0–53.0, n = 11) and 0 (0–40.3; n = 28) at troughs 1 and 2, respectively.

In the low-dose subgroup, the median (IQR) trough PCs (ng/mL) of rivaroxaban users were 25.1 (0–50.2; n = 2) and 0 (0–26.5; n = 12) at troughs 1 and 2, respectively. The trough median (IQR) PCs (ng/mL) of apixaban users were 80.9 (48.7–95.5; n = 6) and 88.4 (65.7–99.2; n = 20) at troughs 1 and 2, respectively. The median (IQR) trough PCs (ng/mL) of edoxaban users were 21.8 (0–38.2; n = 13) and 15.4 (0–41.5; n = 46) at troughs 1 and 2, respectively. The trough drug PCs of the standard- and low-dose subgroups of apixaban were significantly higher than those of rivaroxaban and edoxaban (*p* < 0.01; Figure 2). Peak 1 and trough 2 had the highest number of samples for standard and reduced doses.

### 2.3. Alterations in Peak PCs over Time Among Three DOACs (Figure 2 and Figure 3)

As primary data, we examined the number and incidence of patients with the peak 1 drug PCs above each drug’s cut-off level for predicting bleeding events and their background characteristics (age, BW, and Cr-Cl). Overall, nine (9/57; 15.8%), one (1/29; 3.4%), and three (3/36; 8.3%) standard-dose users of rivaroxaban, apixaban, and edoxaban, respectively, had peak 1 drug PCs higher than the cut-off level. However, no difference in the incidence and background characteristics was found between those with peak 1 drug PCs level higher than the cutoff level and those without among these three FXa inhibitors.

**Figure 3 pharmaceuticals-17-01431-f003:**
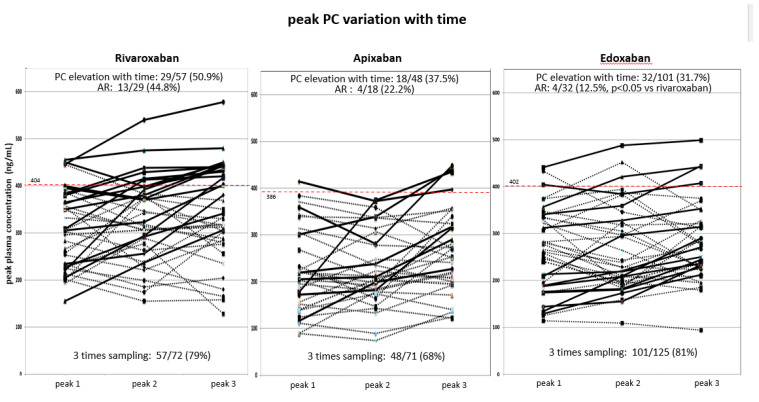
Alteration in peak PCs over time in three FXa inhibitors. Bold solid lines: patients with PCs higher at peak 3 than at peak 1 and/or 2 over time.

Then, the alterations in peak PCs over time in patients with three samples taken for standard- and low-dose use (rivaroxaban, n = 57; apixaban, n = 48; and edoxaban, n = 101) were evaluated. The number of patients with drug PC elevation over time (PC higher in peak 3 than in peak 1 and/or 2) was 29/57 (50.9%) for rivaroxaban users and 18/48 (37.5%, ns vs. rivaroxaban) for apixaban; however, its incidence was lower for edoxaban (32/101; 31.7%, *p* < 0.05 vs. rivaroxaban by chi-square test). Furthermore, among those with high drug PCs over time, the number of patients with drug PCs higher than the cut-off level in peak 3 (accumulation rate) were 13/29 for rivaroxaban (44.8%; standard-dose subgroup, n = 12; low-dose subgroup, n = 1), 4/18 for apixaban (22.2%, all standard dose), and 4/32 for edoxaban (12.5%, standard-dose subgroup, n = 3; low-dose subgroup, n = 1, *p* < 0.05 vs. rivaroxaban). The incidence of drug accumulation was low with edoxaban. In both the rivaroxaban and edoxaban users, the BW decreased in those with peak PCs higher than the cut-off level in peak 3 sampling than in those with the lower one (rivaroxaban, 62 ±11 kg; n = 13 vs. 75 ± 19 kg; n = 37, *p* < 0.05, edoxaban; 63 ± 2 kg; n = 4 vs. 75 ± 9 kg; n = 8, *p* < 0.05); however, the difference may be not conclusive because of the small number of patients.

### 2.4. Bleeding and Thromboembolic Events Among DOAC Users (Figure 1)

Bleeding events occurred in four rivaroxaban users (two each receiving standard and low doses; major in patients R1, R2, and R3 and minor in patient R4; indicated with red numbers), two apixaban users (all receiving standard doses; both major in patients A1 and A2; indicated with red numbers), and four edoxaban users (all receiving standard doses; major in patients E1, E2, and E4 and non-major in patient E3; indicated with red numbers).

Among these 10 patients who experienced bleeding events, eight had peak drug PCs exceeding each drug’s cut-off levels for predicting bleeding events caused by drug overdose based on analysis and these events resulted in overdose based on PC measurement. However, as patients R4 and E4 had drug PCs below the cut-off, bleeding in these two patients was assumed to not be drug-induced (R4, subcutaneous contact bleeding; E4, gastric bleeding due to ulcerated lesion). Meanwhile, no thromboembolic events occurred in this study.

### 2.5. Other Coagulation Biomarkers (Table 2, Figure 4 Top and Bottom)

For the fibrinogen (FBG) levels (normal range: 200–400 mg/dL), to assess thrombin activity, the integrated peak data (peak FBG) were obtained from sampling peaks 1–3 and the integrated trough data (tough FBG) from sampling troughs 1 and 2 from all patients. In the standard- and low-dose subgroups for rivaroxaban, the integrated peak and trough FBG levels (mg/dL) were 278.9 ± 69.5 (n = 156) and 283.7 ± 80.6 (n = 51), and 283.2 ± 58.4 (n = 39) and 289.6 ± 70.4 (n = 14), respectively. In the standard- and low-dose subgroups for apixaban, the integrated peak and trough FBG levels (mg/dL) were 284.4 ± 67.6 (n = 103) and 283.5 ± 63.0 (n = 35), and 301.5 ± 63.0 (n = 77) and 316.9 ± 94.5 (n = 26), respectively. In the standard- and low-dose subgroups for edoxaban, the integrated peak and trough FBG levels (mg/dL) were 300.4 ± 65.9 (n = 118) and 300.6 ± 55.1 (n = 39), and 294.1 ± 71.8 mg/dL (n = 222) and 301.4 ± 62.7 (n = 59), respectively. This indicates that among all three DOACs, the FBG levels at the peak and trough did not significantly differ, regardless of the dose. However, some patients had high FBG levels of >500 mg/dL during the peak drug PC or both during the peak and trough drug PC. Nevertheless, these abnormalities were not related to thrombogenesis.

**Figure 4 pharmaceuticals-17-01431-f004:**
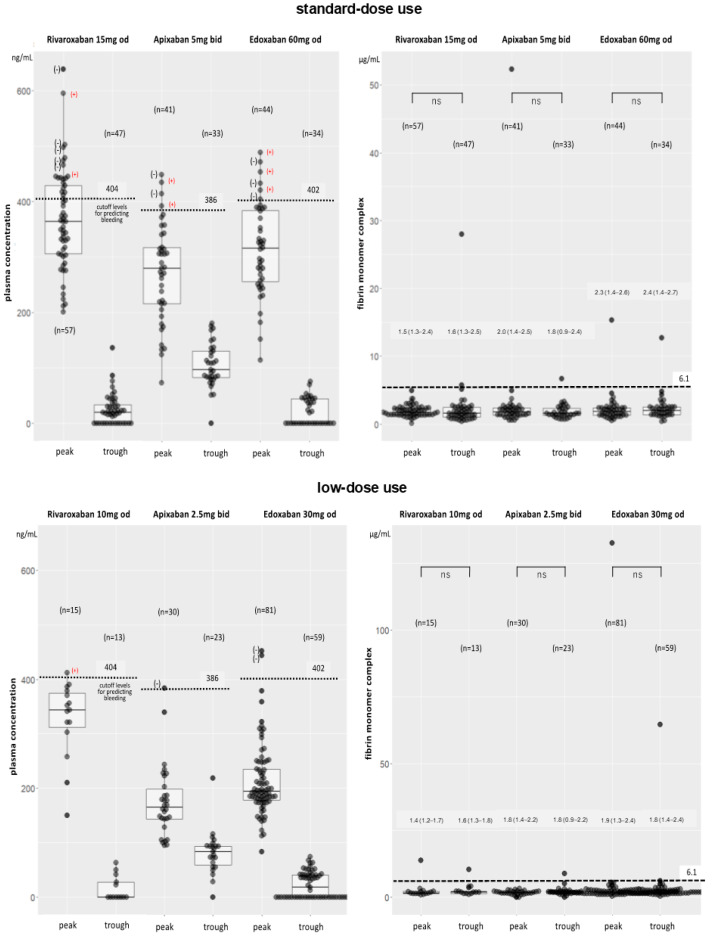
Relationship between drug plasma concentration (PC) (**left**) and fibrin monomer complex (FMC) (**right**) levels at peak and trough times in patients receiving standard- (**top**) and low- (**bottom**) doses of FXa inhibitors. (+): patients with bleeding events. Numbers for FMC indicate median (25th–75th percentile) values for each time. Numbers above dotted lines (**left**) indicate each cut-off PC level. Dotted lines: cut-off levels for predicting bleeding events in each DOAC (**left**), and normal FMC range (<6.1 µg/mL) (**right**).

**Table 2 pharmaceuticals-17-01431-t002:** Risk factors related to bleeding events by multiple logistic regression analysis in 3 FXa inhibitors.

Drug	Rivaroxaban					Apixaban					Edoxaban				
	Bleedingevents					Bleedingevents					Bleedingevents				
	(+)	(−)	Odd Ratio	95%CI	Wald *p* Value	(+)	(−)	Odd Ratio	95%CI	Wald *p* Value	(+)	(−)	Odd Ratio	95%CI	Wald *p* Value
	(n = 4)	(n = 38)				(n = 2)	(n = 54)				(n = 4)	(n = 89)			
Age (yrs)	71 ± 7	66 ± 12	0.386	0.02 to 6.57	0.510	77 ± 13	76 ± 8	1.713	<0.001 to >999.999	0.911	77 ± 6	73 ± 9	0.429	<0.001 to 702.61	0.823
BW (kg)	58 ± 5	75 ± 19	1.921	0.29 to 12.77	0.500	68 ± 8	59 ± 10	1.648	0.004 to 618.37	0.869	71 ± 9	62 ± 14	1.575	0.20 to 12.35	0.665
CrCl (mL/min)	65 ± 17	88 ± 36	0.449	0.06 to 3.50	0.444	62 ± 7	56 ± 20	1.065	0.004 to 276.68	0.982	63 ± 8	63 ± 26	0.702	0.02 to 24.41	0.845
peakPC (ng/mL)	400 ± 180	343 ± 98	0.973	0.90 to 1.05	0.486	428 ± 19	238 ± 91	1.080	0.39 to 3.02	0.883	458 ± 27	259 ± 83	1.199	0.36 to 3.97	0.766
troughPC (ng/mL)	22 ± 43	29 ± 28	0.946	0.85 to 1.05	0.300	159 ± 31	97 ± 40	1.022	0.71 to 1.46	0.906	59 ± 16	18 ± 21	1.061	0.77 to 1.46	0.718
peakFBG (mg/dL)	341 ± 73	287 ± 68	1.096	0.96 to 1.26	0.188	297 ± 79	294 ± 71	1.037	0.50 to 2.13	0.922	338 ± 91	290 ± 73	0.988	0.70 to 1.39	0.947
troughFBG (mg/dL)	277 ± 50	273 ± 61	0.976	0.89 to 1.07	0.613	273 ± 80	301 ± 80	0.984	0.39 to 2.51	0.973	367 ± 31	298 ± 57	1.136	0.64 to 2.02	0.662
peakFMC (μg/mL)	1.3 ± 0.3	5.5 ± 19.6	0.007	<0.001 to >999.999	0.604	0.9 ± 1.0	1.8 ± 0.8	0.001	<0.001 to >999.999	0.741	1.2 ± 0.5	2.0 ± 1.5	0.206	<0.001 to >999.999	0.855
troughFMC (μg/mL)	1.7 ± 1.3	3.2 ± 4.7	0.003	<0.001 to 312.72	0.326	1.1 ± 0.4	2.1 ± 1.4	0.087	<0.001 to >999.999	0.901	2.5 ± 1.3	2.2 ± 1.5	12.270	<0.001 to >999.999	0.810

FBG: fibrinogen; FMC: soluble fibrin monomer complex; PC: plasma concentration. Wald *p* values mean *p* values for Wald chi-square.

Regarding soluble fibrin monomer complex (FMC) levels (normal range, <6.1 µg/mL) being a coagulation biomarker, the integrated peak data (peak FMC) were obtained from sampling of peaks 1–3 and the integrated trough data (trough FMC) from troughs 1 and 2 samples for all patients. In the standard- and low-dose subgroups of rivaroxaban, the integrated peak and trough FMC levels (µg/mL) were 3.1 ± 11.4 (n = 156) and 4.1 ± 13.6 (n = 51), and 3.3 ± 5.6 (n = 39) and 2.6 ± 2.4 (n = 14), respectively. In the standard- and low-dose subgroups of apixaban, the integrated peak and trough FMC levels (µg/mL) were 2.3 ± 4.9 (n = 108) and 1.9 ± 1.1 (n = 35), and 2.0 ± 1.1 (n = 77) and 2.0 ± 1.6 (n = 33), respectively. In the standard- and low-dose subgroups of edoxaban, the integrated peak and trough FMC levels (µg/mL) were 2.1 ± 1.9 (n = 118) and 2.3 ± 1.9 (n = 39), and 4.2 ± 16.2 (n = 222) and 3.2 ± 8.2 (n = 59), respectively. Among all three DOACs, the FMC levels at peaks and troughs did not significantly differ regardless of the dose. However, as shown by the FBG levels, some patients only had FMC levels of >100 µg/mL at the peak or both at the peak and trough. Additionally, large standard deviations were observed among rivaroxaban and edoxaban users. However, these abnormalities were assumed to not be caused by thrombogenesis. The integrated FBG and FMC data between patients with and without bleeding events among these three DOACs are shown in Table 2.

Then, the relationship between drug PCs and FMC levels at peak and trough times was compared for three FXa inhibitors. For these comparisons, as peak FMC data, the FMC data at the highest peak PC level from sampling at peaks 1, 2, and 3 were selected, and as trough FMC, the FMC data at the lowest PC level from sampling at trough 1 or 2 in both of standard- and low-dose uses. Although the drug PCs varied widely from peak to trough, FMC levels did not significantly vary regardless of the dose or frequency of drug administration. Notably, FMC levels were maintained even in patients on the once-daily drugs, rivaroxaban and edoxaban (Figure 4).

### 2.6. Indexes for Predicting Bleeding Events (Table 2)

Although bleeding occurred too infrequently, which indexes were related to the bleeding events were evaluated with logistic regression analyses. The following indexes were used: age, BW, Cr-Cl, peak and trough PCs, peak and trough FBG, and peak and trough FMC. Given some deficits in the sampling, particularly in the trough data, we conducted these with the patient numbers shown in Table 2. However, no factors were related to bleeding events. Unsurprisingly, we could not also perform receiver-operating characteristic analysis to determine the cut-off level for predicting bleeding events.

## 3. Discussion

This multicenter registry study (the SET DOAC registry; UMIN000036769) was performed to evaluate the efficacy and safety of three FXa inhibitors (rivaroxaban, apixaban, and edoxaban) coupled with the monitoring of drug PCs and coagulation biomarkers at peak and trough levels. These three DOACs demonstrated comparable antithrombotic effects, even at trough levels, regardless of the frequency of administration. However, rivaroxaban tended to accumulate over time when compared with edoxaban.

The concern with once-daily DOACs is that their effects may be reduced when drug PCs are at their trough [10,11]. Meanwhile, twice-daily DOACs have two peaks, and their trough PC levels are not as low as those of once-daily DOACs [10]. Therefore, when considering DOACs for the secondary prevention of stroke, physicians tend to select twice-daily DOACs. Unfortunately, the number of the study’s patients was too small to determine the anticoagulant effect of the three DOACs based on their PCs alone. However, levels of coagulant biomarkers used in this study (FBG and FMC) remained low even during trough PCs in patients taking once-daily DOACs. This indicates that the risk of thrombotic events on using once-daily DOACs is low.

Several large randomized trials have reported the efficacy and the safety of the three FXa inhibitors [2,3,5,7]. When compared with warfarin, rivaroxaban and edoxaban were noninferior, whereas apixaban was superior, in preventing stroke or systemic embolism. Regarding bleeding events, apixaban and edoxaban resulted in a lower bleeding incidence than warfarin; however, no significant difference in the incidence of major bleeding was found between rivaroxaban and warfarin [2]. Meanwhile, two studies have reported that apixaban and edoxaban were associated with lower rates of ischemic stroke than rivaroxaban [2,12]. In contrast, a study reported that the incidence of ischemic stroke was not significantly different among the three DOACs and dabigatran [13]. Additionally, studies comparing once- with twice-daily DOACs revealed that twice-daily DOACs might have a more balanced risk–benefit profile between thromboembolism and bleeding events [14,15]. However, recent studies have reported that there is no clear difference between once- and twice-daily DOACs in terms of adverse outcomes, and the incidence of adverse outcomes is higher in patients with low adherence than in those with high adherence, regardless of the dosing frequency [16,17].

Regarding the activity of coagulation biomarkers at trough levels, Shinohara et al. examined the drug PCs, D-dimer levels, and prothrombin fragment 1 + 2, which represented coagulation biomarkers, in patients who temporarily discontinued edoxaban for catheter ablation [18]. They reported that the drug PC further decreased 24 h after discontinuing edoxaban; however, the plasma levels of the prothrombin fragment remained unchanged even after 30 h. In the present study, which evaluated FBG and FMC levels as coagulation biomarkers, FMC was used as an alternative for assessing thrombosis, in addition to FBG, because it reflects thrombin activity and can be detected earlier than D-dimer [19]. Therefore, as regards the level of coagulation biomarkers, no clear difference was noted between once- and twice-daily DOACs. Currently, the half-life of FMC remains unclear. The finding of the lack of a difference in FMC levels between peak and trough values may be inconclusive.

In this study, we could not assess safety as regards bleeding in DOACs, given the low incidence of bleeding. However, in routine practice, rivaroxaban is associated with a higher bleeding incidence than apixaban and edoxaban [2,3,4,5,6,7]. Additionally, in patients with an extreme BW, a practical guide to DOAC use indicates that special care is needed in particular for individuals with a BW of ≤60 kg, in whom low doses of apixaban and edoxaban should be administered [1,8]. Also, the EHRA guide indicates considering PC monitoring when prescribing rivaroxaban or dabigatran in patients with a BW of <60 kg. Furthermore, in a preliminary study from our institute preceding the SET DOAC registry, edoxaban had a more favorable safety profile than rivaroxaban in terms of PC monitoring (Hokusetsu DOAC study) [9]. Regarding bleeding events with rivaroxaban, the results showed that the drug accumulation may be related to bleeding.

This study had some limitations. Although this study was conducted at several institutions, the number of enrolled patients and the incidence of bleeding events were very low to draw definitive conclusions about the risk of bleeding events among patients taking these three drugs. Although we could determine variations in PCs over time among the three DOACs, this result may be affected by the small sample size. In this study, the Biophen DiXaI kit that employed the chromogenic anti-FXa assay was used. Concerning drug–drug interaction affected bleeding events, we showed the data of antiplatelet drugs in Table 1, and a few patients took non-steroid anti-inflammatory drugs without relation to bleeding events. We used the cut-off PC levels obtained from our previous study. Considering that few kits currently provide a chromogenic assay, the cut-off PC levels for predicting bleeding events may vary depending on the kit used. Therefore, another multicenter study may be needed to confirm the results.

## 4. Methods

### 4.1. Study Design and Participants

This observational study included outpatients with NVAF from nine hospitals, including Hokusetsu General Hospital, who were seen between November 2018 and December 2021. We planned to enroll 300 patients for each drug; however, because of the COVID-19 pandemic, only 268 outpatients were enrolled.

The Cr-Cl was calculated using the Cockcroft–Gault formula adjusted for age, BW, serum creatinine level, and sex. Additionally, the CHADS_2_ scores were calculated immediately before DOAC treatment initiation based on the presence of heart failure, HT, an age ≥ 75 years, diabetes mellitus, and stroke history.

The Japanese standard on-label dose for rivaroxaban is 15 mg once-daily (od); the recommended on-label low dose of 10 mg od is used for patients with a Cr-Cl of <50 mL/min [20]. The standard dose of apixaban is 5 mg twice daily (bid); however, 2.5 mg bid is the recommended on-label low dose for patients meeting ≥2 criteria for dose reduction (S-Cr of ≥1.5 mg/dL, BW of ≤60 kg, and age of ≥80 years). The standard on-label dose of edoxaban is 60 mg od, whereas 30 mg od is the recommended on-label low dose for patients with a BW of ≤60 kg, Cr-Cl of <50 mL/min, or for those concomitantly using P-glycoprotein inhibitors. Therefore, DOAC doses for patients were selected based on the abovementioned dose recommendations.

Patients who were new users and expressed a desire to participate in this trial, which required blood sampling on several occasions, and underwent at least the first blood sampling (peak 1), were eligible for this study; however, patients who were treated with off-label low doses were not included. All patients provided written informed consent, and this study was approved by the ethical review committee at each hospital.

### 4.2. Measured Parameters

FBG (with Thrombocheck Fib [L]^®^, Sysmex, Kobe, Japan) and FMC (with Auto LIA FM^®^: Sysmex, Kobe, Japan) were measured using an automated blood coagulation analyzer (CS-5100; Sysmex, Kobe, Japan). Under a steady state of at least 4 weeks after the initiation of each DOAC, drug PCs in each DOACs group were measured via the anti-FXa-derived indirect method using the Biophen^®^ DiXaI kit (Hyphen Biomed, Neuville Sur Oise, France) [21] with the same protocol as our previous studies [9,22]. This kit uses a chromogenic method based on inhibiting a constant and excess quantity of FXa by the drug being assayed, with a calibrator and control for each drug.

### 4.3. Blood Sampling (Figure 2)

After the plasma was separated from the collected blood samples under 4 °C cooling and plasma samples were frozen at each hospital, these frozen samples were analyzed at the Reagent Engineering (TS) division of Sysmex Corporation.

Blood samples were collected 3 h after drug administration to measure the peak drug PC and immediately before drug administration to measure the trough drug PC [1,9,22], as described previously. We measured PC and two coagulation parameters, FBG and FMC, at the same time with the same samples. To evaluate variations in PC levels over time, the peak drug PC was also measured thrice and trough drug PC twice over 6–12 months of continuous DOAC administration (peak 1, 1 month later; peak 2, 3–4 months later; peak 3, 6–12 months later; trough 1, 1 month later; trough 2, 3–4 months later).

At peak 1, the incidence of the drug PC exceeding the cut-off level for predicting the bleeding events was evaluated. Each cut-off level was based on our previous investigations (rivaroxaban, 404 ng/mL; apixaban, 386 ng/mL; edoxaban, 402 ng/mL) [9,22]. Furthermore, the variations in peak drug PC over time were examined by measuring peak 3 and comparing it with peak 1 and/or peak 2 (for evaluating the tendency toward drug accumulation).

### 4.4. Bleeding and Thromboembolic Events in DOAC Users

According to the definition by the International Society on Thrombosis and Haemostasis, major bleeding is a clinically overt bleeding associated with a ≥2.0 g/dL decrease in the hemoglobin level, a transfusion requirement of ≥2 U of packed red cells or whole blood, the involvement of a critical site, or a fatal outcome [23]. Non-major clinically relevant bleeding was defined as clinically overt bleeding that did not meet the criteria for major bleeding, but required medical intervention, unscheduled consultation with a physician, or the temporary discontinuation of study treatment, and resulted in pain or the impairment of daily activities; overt bleeding episodes that did not meet the criteria for major or non-major clinically relevant bleeding were classified as minor bleeding [24]. Thromboembolic events were also defined using the same definition as the outcome and endpoint components in the J-ROCKET AF study [24]. Therefore, additional blood samples were taken from patients who experienced bleeding events at the time of bleeding, and follow-up was interrupted.

All study patients were followed up in each hospital. Blood samples were collected from patients who experienced a bleeding event or neurological symptoms, and the next course of treatment, such as drug interruption or further therapeutic procedures, was determined at each hospital. The follow-up outcomes were also recorded by reviewing outpatient records or making phone calls to the general physician and patients. Data collection was discontinued on 30 December 2021.

### 4.5. Statistical Analyses

A physician (MS) and statisticians (Hajime Yamakage, M Eng, et al., Satista, Co., Ltd., Kyoto, Japan) performed all statistical analyses using SPSS Statistics version 24 (IBM Corp., New York, NY, USA). Statistical analyses were performed based on a 5% level of significance. The Wilcoxon two-sample test was used to compare the continuous variables. Logistic regression analyses were performed to determine the risk factors related to bleeding events. Data are expressed as the mean ± SD or median and 25th–75th percentile.

## 5. Conclusions

Coagulant activity, as shown by FMC, was suppressed throughout the day even if the drug PC levels of once-daily DOACs declined to nearly zero at the trough level, indicating that the risk of thrombotic events remains low even for these drugs. Compared with edoxaban, rivaroxaban results in drug accumulation, which may increase drug PCs notably higher than the cut-off for bleeding events over time.

## Data Availability

Data is contained within the article.

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
