# Peer review of "Efficacy and Safety of Rivaroxaban, Apixaban, and Edoxaban for Nonvalvular Atrial Fibrillation Based on Blood Coagulation Activity and Drug Plasma Concentration: SETtsu and North Osaka Multicenter Direct Oral AntiCoagulant (SET DOAC) Registry"

_pharmaceuticals, 2024, doi:10.3390/ph17111431_

Round 1

Reviewer 1 Report

Comments and Suggestions for Authors

The study aimed to evaluate the efficacy and safety of Rivaroxaban, Apixaban, and Edoxaban in a real-world clinical setting, focusing on their impact on blood coagulation activity and plasma concentration levels.

    • The study reported differences in the pharmacokinetics and pharmacodynamics of the three anticoagulants.
    • Efficacy was measured through the reduction of thromboembolic events, while safety was assessed by monitoring bleeding complications.
    • Results indicated that all three DOACs were effective in reducing the risk of stroke and systemic embolism in NVAF patients, with varying degrees of safety profiles.

INTRODUCTION

It remains to describe the drugs used at work. It is necessary to know a little about them.

METHODS

Converting figure 1 into a table would make the information it contains better appreciated.

RESULTS

Figure 2 is complicated for the reader, it is suggested to improve the quality, the variables to be highlighted in the study are not observed. Create a more appropriate way to present the data.

With the data presented, a principal component analysis could be performed to observe if there are differences between the drugs evaluated with the different doses. Even do a PLS.  

Discussion

Not comment.

Conclusions

Not comment.

Author Response

Comments 1.In Introduction: It remains to describe the drugs used at work. It is necessary to know a little about them.

Response 1: These 3 drugs studied in this investigation are factor Xa inhibitors (FXa-I) and 4 drugs added dabigatran being thrombin inhibitor are named direct oral anticoagulants (DOAC). Especially, these FXa-Is are widespread used as oral anticoagulants. Is there necessary to describe them?

Comments 2.Converting Figure 1 into a Table: 

Response 2: We would like to keep Figure 1 in the present form but we noticed to clarify more the flow of this investigation by your suggestions. So, we added the numbering with marking of # and also used these numbering in Figure 2.

Comments 3.Suggestion to improve the quality in Figure 2:

Response 3: First of all, Figure 2 was rotated vertically and was made in 3 layers (top, middle and bottom). Also, we added the timing of blood sampling with #2~#6 being the number in the flow chart. Also, we clarified the profiles of patients with bleeding events with red numbering and added the data of PC levels in Figure 2 only in patients with high PC levels.

Comments 4.Is it needed to perform principal component analysis?:

Response 4: We appreciated this comment. First, by this analysis we consider that we may catch which indexes is related to bleeding events among 3 drugs and two doses. However, the incidence of bleeding was too low. Therefore, we re-evaluated logistic regression analysis to find the parameters related to the bleeding events and showed the re-evaluated data in new Table 2 and section: Results 2.6. However, we could not find it.

Reviewer 2 Report

Comments and Suggestions for Authors

Suwa et al. sought to investigate pharmacodynamics and pharmaokinetics of factor Xa (FXa) inhibitors as well as their safety and effectiveness for predicting bleeding and thromboembolic complications, respectively. The authors found that although drug plasma concentration (PC) levels of once-daily FXa inhibitors widely varied from peak to trough, fibrin monomer complex (FMC) levels maintained within the normal range without daily variations. Simultaneously, the risk of thrombotic events with once-daily FXa inhibitors was low and equivalent to twice-daily one. Study results are interesting; however, some issues require explanation.

The results section of the abstract require data supporting conclusions rather than descriptive information.

Inclusion and exclusion criteria to this study are missing.

Please provide independent predictors of PC of FXa inhibitors as well as of FMC. Are they similar?

Please add ROC curves for PC of FXa and FMC for prediction of bleeding complications.

Although mentioned in statistical section, multivariable analysis for predicting of bleeding complications is not easy to understand. Please provide results of multivariable regression, optimally in the table.

Author Response

Comments 1. In the results section of the abstract, the data supporting conclusions is required:                                                        Response 1: We had two major important results in this manuscript; 1. PC elevation with time, especially in rivaroxaban use, 2 FMC being a coagulation biomarker remained within the normal range through the course although PC levels widely varied from peak to trough timing. So, as suggested from Reviewer 2 we added the following short results in the abstract; (peak PC elevation over time, 50.9% in rivaroxaban and 31.7% in edoxaban, p< 0.05). However, we could not add the data concerning the relation between PC variation and FMC because the data were complicated on showing them.

Comments 2.Inclusion and exclusion criteria are missing.

Response 2: We already commented the exclusion of 21 patients from total 289 in the Results section 2.1.: Patients characteristics.   However, in the section of Methods, 4.1., we newly added the following sentence. [Patients who expressed the participation in this clinical trial that needed blood sampling on several occasions were included; however, patients who were treated with off-label low doses were not included. Additionally, all patients provided written informed consent, and this study was approved by the ethical review committee at each hospital.]

Comments 3.Please provide independent predictors of PC of FXa inhibitors as well as FMC. Are they similar?                                                     

Response 3: Currently, PC levels in FXa inhibitors indicate drug activity and varied widely from peak to trough timing, similar to prothrombin time. On the contrary, fibrin monomer complex (FMC) reflects thrombin activity (In the abstract, we added the following sentence: reflecting thrombin activity) and is similar to D-dimer. Currently, plasma concentration of DOAC varied widely throughout the day, especially in once-daily drug. Therefore, twice-daily drug may have more beneficial to prevent thromboembolic events. Thus, we evaluated whether there were some distinctions between once or twice-daily drug with PC level and FMC data. Also, we added the data obtained from logistic regression analysis contained PC and FMC data in Table 2.

Comments 4.Please add ROC curves for PC of FXa inhibitors and FMC for prediction of bleeding complication.                                                     

Response 4:  In this study, the incidence of bleeding events was too low. Therefore, we could not take cut-off level for predicting bleeding events using ROC analysis from this study. However, we had the similar data in the former study with the same protocol (Circulation Journal 2019 and Circulation Reports 2023) and used the previous data in this study. FMC is an index for not predicting bleeding but shows coagulation activity. In this study, FMC suppressed throughout the day and did not raise at the trough time on FXa inhibitors dosing. Also, as shown in the next comment, we did not evaluate the cut-off data using ROC curves concerning FMC because there was no statistical significance in FMC data between patients with and without bleeding events.                                           

Comments 5.Please provide the results of multivariate regression analysis.            

Response 5: We would like to apologize that Table 1 in pre-revised manuscript did not show the precise data of logistic regression analysis. So, we re-evaluated this analysis and showed them in Table 2 and Results 2.6. in the text. Because the incidence of bleeding events was very low, on re-evaluated data of logistic regression analysis we could not find the significance between patients with and without bleeding events, shown in Results section 2.6. So, we did not perform ROC analysis. However, we used the former data obtained with the same protocol from our institute (Circulation Journal 2019 and Circulation Reports 2023).

Reviewer 3 Report

Comments and Suggestions for Authors

 The safety and effectiveness of factor-Xa (FXa) inhibitors for nonvalvular atrial fibrillation (NVAF) remains inconclusive. The authors investigated the current dosing of rivaroxaban, apixaban, and edoxaban by monitoring drug plasma concentration (PC) and coagulation activity.

Suggested minor corrections:

1. In the introductory part, the authors could provide a scheme of anti-thrombotic action in which they would indicate the location of the investigated molecules.

2. The authors should write a few sentences about rivaroxaban, apixaban and edoxaban in the introductory part.

3. The authors should also explain the condition of nonvalvular atrial fibrillation (NVAF) in a little more detail.

4. Figure 2 and Figure 3 need to be rotated by 90 degrees or given some other arrangement of graphs.

Author Response

Comments 1. In the introductory part, the authors could provide a scheme of anti-thrombotic action in which they would indicate the location of the investigated molecules.                                                       Response 1: We understand your recommendation. However, these three Factor Xa inhibitors have the same action as the drugs name express their drug characters themselves. Also, as our manuscript is not a review article but an investigative one, we did not think that there is a need to show the scheme.

Comments 2.The authors should write a few sentences about three these drugs in the introductory part.                                                                                                                                                        Response 2: We had already commented that among 3 Factor Xa inhibitors rivaroxaban has a higher bleeding incidence compared with other two in the introduciton. On this revision, we added the following phrase; [So, we may need to recognize that these three Factor-Xa inhibitors do not always have the same safety profile.]   

Comments 3.The authors should also explain the condition of nonvalvular atrial fibrillation in a little more detail.                                                                                                                                           Response 3: We showed the patients profile in new Table 1 and added the data concerning the pattern of atrial fibrillation among 3 drug groups with two dosing type.  

Comments 4.Figures 2 and 3 need to be rotated by 90 degrees or given some other arrangement of graphs.                                                                                                                                                   Response 4: We appreciated your comments. So, we performed the arrangement.

Round 2

Reviewer 1 Report

Comments and Suggestions for Authors

The work has improved with the corrections requested from the authors. I suggest the job be accepted. I have no comments for the authors.  

Author Response

I re-made the manuscript according to the comment.

Reviewer 2 Report

Comments and Suggestions for Authors

I have no more comments.

Author Response

I corrected the manuscript according to the reviewer's commnet.
